# Metabolically flexible microorganisms rapidly establish glacial foreland ecosystems

Francesco Ricci [1,2,11], Sean K. Bay [1,3,11], Philipp A. Nauer[4], Wei Wen Wong[4], Gaofeng Ni [1], Luis Jimenez[1], Thanavit Jirapanjawat[1], Pok Man Leung [1,2], James A. Bradley [5,6], Vera M. Eate[4], Montgomery Hall [1], Astrid K. M. Stubbusch [1], Beatriz Fernández-Marín[7], Asunción de los Ríos [8], Perran L. M. Cook [4], Martin H. Schroth[9], Eleonora Chiri[1,10] ✉ & Chris Greening [1,2,10] ✉

An overriding question in ecology is how new ecosystems form. This question can be tested by studying colonisation of environments with little to no pre-existing life. Here, we investigated the functional basis of microbial colonisation in the forelands of a maritime Antarctic and an alpine Swiss retreating glacier, by integrating quantitative ecology, metagenomics, and biogeochemical measurements. Habitat generalists and opportunists rapidly colonise both forelands and persist across soil decadal chronosequences serving as proxies for temporal community dynamics. These microbes are metabolically flexible chemotrophic aerobes that overcome oligotrophic conditions by using organic and inorganic compounds, including atmospheric trace gases and sulphur substrates, for energy and carbon acquisition. They co-exist with metabolically flexible early-colonising opportunists and metabolically restricted later-colonising specialists, including Cyanobacteria, ammonia-oxidising archaea, and obligate predatory and symbiotic bacteria, that exhibit narrower habitat distributions. Analysis of 589 species-level metagenome-assembled genomes reveals early colonisation by generalists and opportunists is strongly associated with metabolic flexibility. Field- and laboratory-based biogeochemical measurements reveal the activity of metabolically flexible microbes rapidly commenced in the forelands. Altogether, these findings suggest primary succession in glacial foreland soils is driven by self-sufficient metabolically flexible bacteria that mediate chemosynthetic primary production and likely provide a more hospitable environment for subsequent colonisation.

Environments with little or no preexisting life can serve as natural laboratories to understand how complex communities establish and thrive[1–5]. Pioneer microorganisms (i.e., primary colonisers) establish such systems, performing key roles such as primary production, nutrient recycling, weathering, and detoxification, thereby increasing habitability and supporting secondary colonisation of diverse species[6–9]. Once microbial communities are established, cryptogams, vascular plants, and animals can sometimes colonise[9]. Over time, the assembly of microbial communities is governed by the interplay between stochastic (including dispersal from air and water flow) and deterministic (including selection due to biotic and abiotic factors) processes[10,11]. As pioneer ecosystems are typically carbon- and energy-limited, primary producers are particularly important ecosystem engineers[12]. Accordingly, most classical studies on pioneer

---

microorganisms have focused on photosynthetic microbes such as Cyanobacteria[13–15]. Primary producers also include chemosynthetic microbes that use lithic substrates (e.g., $NH_4^+$, $Fe^{2+}$, and $S^{2-}$)[16,17] and trace gases (e.g., $H_2$, CO, and $CH_4$)[18,19] to drive aerobic respiration and carbon fixation. While photosynthetic microbes are often metabolically constrained by light and water availability, many chemosynthetic microbes are highly flexible, for example, alternating between inorganic and organic substrates in response to fluctuating environmental conditions and resource availability, and thus may be able to better tolerate the harsh and variable conditions of many pioneer ecosystems[20–22]. Our discovery of aerotrophy, i.e., the use of atmospheric trace gases to drive aerobic respiration and carbon fixation, particularly in oligotrophic environments (e.g., Antarctic deserts)[23–25], suggests that the atmosphere may be a particularly dependable energy supply for microbial pioneers that may enable continuous primary production during ecosystem establishment. To date, multiple studies have investigated compositional changes and ecological dynamics of microbial communities during colonisation and succession[11,26]. However, the functional traits and biogeochemical processes that enable microbes to colonise remain unresolved.

Glacial forelands are ideal systems to study ecosystem formation[27]. The retreat of land-terminating glaciers, as accelerated by anthropogenic global warming, produces forelands that are gradually colonised. The primary succession of these forelands can be studied using a chronosequence approach (assuming space-for-time substitution[28]), where sites of increasing distance away from the retreating glacier front represent increasing soil ages[9,29–32]. Recently deglaciated lands are harsh environments for microbial colonisers, typically characterised by low organic carbon and nitrogen, high irradiance, and large temperature fluctuations, which impose strong selective pressures on colonisation[9,30,33]. These pressures can be especially extreme in places that are already at the fringe of life, such as terrestrial Antarctica. In such systems, early colonisation processes involve a trade-off between surviving these physicochemical pressures and exploiting available resources[33]. Initial colonisers may disperse from various sources: for instance, ice-dwelling microbes can persist on deglaciated soil[34], while others can be airborne or transported through hydrologic flow[35,36]. Regardless of their origin, selected microbes rapidly colonise and mediate biogeochemical cycling in newly exposed soil, facilitating secondary colonisation[9,37]. Previous studies have begun to use metagenomics to gain insight into the functional strategies supporting microbial colonisation[30,38,39]. For example, they have suggested that primary colonisers overcome nitrogen limitation in forelands through a multiplicity of strategies, including assimilation of inorganic nitrogen, degradation of organo-nitrogen compounds, and nitrogen fixation[30,33,40]. While certain biogeochemical processes are conserved throughout foreland chronosequences, later successional stages may result in the emergence of other metabolisms, such as methane oxidation and nitrification[19,30,40,41]. Yet despite this emerging knowledge base, we lack a consolidated understanding of the relationship between the ecological dynamics, functional traits, and biogeochemical activities driving microbial colonisation and subsequent succession, including the metabolic breadth of pioneer microbes and roles of different primary production strategies[3,27,32].

Ecological theory, originally developed for plant communities, provides a quantitative framework to examine colonisation and succession dynamics[42,43]. A central theory is that habitat generalists (i.e., organisms that can withstand a wide range of environmental conditions and resource availabilities) are early colonisers, while specialist taxa (i.e., organisms exhibiting narrower habitat preferences and with stricter resource requirements) generally increase during succession[44,45]. This concept, together with trait-based frameworks[42], offers a baseline to examine microbial primary succession from a functional standpoint. However, it is important to note that microbes generally exhibit greater metabolic flexibility than plants and animals, with many able to use multiple carbon and energy sources either simultaneously (mixotrophy) or alternatively[21,46]. Most can also meet energy needs under resource limitation through continuous energy harvesting using trace gases or sunlight (via rhodopsins and photosystems) while in dormant or slow-growing states[47]. Therefore, it is possible that microbial communities do not conform to classic macroecological succession dynamics because of their high degree of flexibility, which could provide an ecological advantage from early to late successional stages[48]. Given these considerations, we hypothesised that the dominant pioneers in glacial forelands would be metabolically flexible bacteria that overcome nutrient limitation by conserving energy and fixing carbon using atmospheric trace gases. In this work, we provide a system-wide understanding of the community and functional dynamics underpinning colonisation of two glacial forelands, by integrating gene- and genome-centric metagenomics with biogeochemical measurements and ecological theory. Our study focused on two contrasting sites. Hurd Glacier on Livingston Island, Antarctica, is a remote maritime glacier, terminating over the Southern Ocean, that forms part of the 10 km² Hurd Peninsula ice cap alongside Johnsons Glacier[49]. Griessfirn Glacier in Switzerland is also small (~5 km²), but it lies at 2400–3300 metres above sea level[50], and is more likely to receive external resource inputs and experience organismal colonisation or invasion due to its central position in the Swiss Alps. Our findings provide a comprehensive understanding of microbial metabolic strategies that enable the colonisation and establishment of ecosystems from their earliest stages of succession to communities that have persisted for over a century.

## Results

### Microbes rapidly and deterministically colonise foreland chronosequences

We sampled surface foreland soils of a maritime Antarctic and an alpine Swiss glacier, covering chronosequences of 21 and 127 years since deglaciation, respectively (Fig. 1a and Supplementary Data 1), as well as depth profiles (from 0 to 50 cm) for the Swiss soils. Geochemical analyses showed that the foreland soils have low organic carbon ($0.14 \pm 0.02\%$ Antarctic, $0.57 \pm 0.24\%$ Swiss) and nitrogen (typically < 0.02%), with low salinity and alkaline pH (Supplementary Data 1). Thus, as anticipated[9], these newly deglaciated soils are highly oligotrophic especially at the Antarctic site and therefore are challenging substrates for microbial colonisation. Yet complex microbial communities inhabited even the most recently deglaciated soils (Fig. 1b). Our study design does not allow tracking of the origins of the microbes present, but potential sources include snow and airborne dispersal, glacier meltwater transport and deposition, and microbes that previously inhabited the subglacial environment that became exposed by ice retreat[34,51,52]. Microbial abundance (16S rRNA gene copy number per gram dry weight) increased on average 8-fold across the Swiss and Antarctic chronosequences (Fig. 1c and Supplementary Data 2). Similarly, microbial richness increased along the chronosequences, with Antarctic soil showing the sharpest increase from recently deglaciated (av. 195 amplicon sequence variants, ASVs; Shannon index 4.33) to more mature soils (691 ASVs, Shannon index 5.71) (Fig. 1d and Supplementary Data 2). Microbial composition of the two sites (based on β-diversity ordinations) were distinct, with strong differentiation by site location ($p = 0.0001$) and soil age ($p = 0.0001$; Fig. 1e and Supplementary Data 2). Following colonisation, we observed an increase in community similarity in later stages of succession (Fig. 1e), as previously observed in Antarctic forelands[31]. These patterns, consistent with previous studies[31,53,54], suggest that foreland microbial communities may respond to common successional dynamics[55].

Communities at both glacial forelands were dominated by common soil phyla, especially Proteobacteria and Actinobacteriota

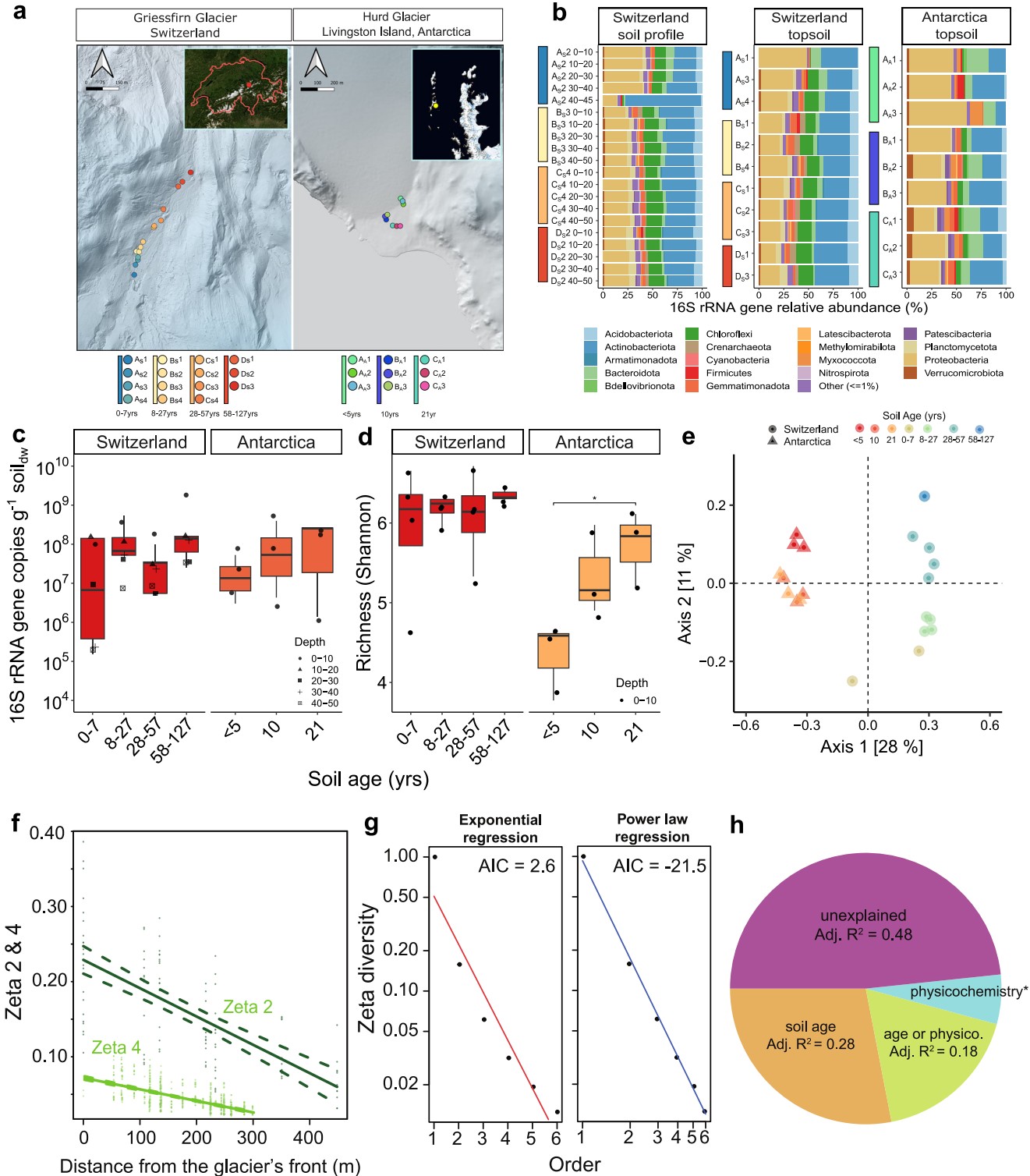

(Fig. 1b), in line with previous studies of receding glaciers[30,56–58]. This was also reflected at finer resolution, with Proteobacteria (e.g., *Devosia*, *Methylotenera*) and Actinobacteriota (e.g., *Pseudarthrobacter*, *Gaiella*) accounting for two thirds of the hundred most abundant ASVs and half of the hundred most widespread ASVs across the two sites (Supplementary Data 2). The Swiss samples also contained two highly abundant archaeal ASVs (av. 1.8% community) that affiliated with the family Nitrososphaeraceae, known to harbour cold-adapted ammonia oxidisers[59]. Diverse photosynthetic Cyanobacteria were also detected, spanning 147 ASVs, though they were at low relative abundance at both

forelands (av. 0.27% Swiss, 0.37% Antarctic; Supplementary Data 2); an exception was one Swiss topsoil (28–57 years old) where a single ASV from the filamentous genus *Tychonema* comprised 2.6% of the community, suggesting these habitat specialists can thrive in certain established soils (Fig. 1b). Concordantly, chlorophyll *a* from Cyanobacteria and microalgae was minimal across the forelands of both glaciers (av. 0.05 μg $g_{soil}^{-1}$ Swiss; 0.14 μg $g_{soil}^{-1}$ Antarctic) but increased progressively with soil age (Supplementary Data 2). Similarly, some methanotrophs (e.g., *Methylomonas* and *Methyloglobulus*) were present but at low occupancy. The Antarctic foreland also had high but

**Fig. 1 | Deterministic processes drive microbial community assembly in glacial forelands. a** Satellite imagery[117] showing the geographic setting of the Swiss and Antarctic glacier forelands studied at the regional level. Sampling design at the local level, with major topographic features, study, scale and sites are shown[118,119]. **b** Stacked bar chart showing phylum-level community structure at the sample level. Vertical-coloured bars show the sample clusters according to soil age. **c** Boxplot showing 16S rRNA gene copy number per gram dry weight ($g_{dw}$) for soils at multiple depths and according to location and soil age ($n = 29$). **d** Boxplot showing microbial community richness (Shannon diversity index) for topsoil (0–10 cm), according to location and soil age ($n = 24$). Significant pairwise differences in soil age are shown (generalised linear model, two-sided contrast with Benjamini-Hochberg correction, $p = 0.02$). Box plots centre line represents the median, the box bounds denote the 25th and 75th percentiles, and the whiskers extend to the most extreme data points

within 1.5 × the interquartile range. **e** Unconstrained Principal Coordinate Analysis (PCoA) showing beta diversity (Bray-Curtis dissimilatory index) of topsoil (0–10 cm) and compositional differences in community structure according to location and soil age. Exact $p$-values and model statistics for a two-sided PERMANOVA are shown in Supplementary Data 2. **f** Zeta distance decay relationship showing community turnover across the Swiss chronosequence ($n = 15$). The number of ASVs shared between two sites ($\zeta_2$) and four sites ($\zeta_4$) with distance are compared. **g** Zeta decline of the Swiss chronosequence modelled by exponential and power law regressions, associated with stochastic and deterministic processes respectively, with Akaike Information Criterion (AIC) scores ($n = 15$). **h** Variation partitioning of zeta diversity across two sites (Zeta orders) of the Swiss chronosequence, explaining variation due to distance and physicochemical factors (*adj. $R^2 = 0.059$; $n = 15$).

variable levels of the obligate predator Bdellovibrionota (av. 1.4%) and obligate symbiont Patescibacteria (av. 1.74%), in support of our previous observations in Antarctic soils[25]. To substantiate these inferences, we calculated specialisation indices for each genus based on the coefficient of variance of their relative abundance across samples as previously described[21,60,61] (Supplementary Data 2). This confirmed the co-existence of habitat generalists with broad occupancies (e.g., *Pseudarthrobacter* and *Gaiella*) alongside habitat specialists with narrow distributions (e.g., *Sericytochromatia* and *Methyloglobulus*) and those with intermediate specialisations (e.g., *Methylotenera* and *Polaromonas*; present in all Antarctic samples but at highly variable abundance; Supplementary Data 2).

Zeta diversity was used to understand the patterns and drivers of community turnover along the Swiss glacial forelands. This new incidence-based metric compares the average number of taxa shared between multiple sites as the number of sites (zeta decline) or distance (zeta decay) increases[62,63]. Zeta diversity declined at a moderate rate, with an average of 23% of ASVs shared between two sites ($\zeta_2$) and 6.6% between four sites ($\zeta_4$; Supplementary Fig. S1), and the coefficient of distance decay was threefold higher for two-way compared to four-way comparisons (Fig. 1f). Together, these zeta diversity and habitat specialisation calculations suggest that most microbes inhabiting these forelands are habitat specialists that undergo rapid turnover, but co-exist with many habitat generalists that persist across the transect and are disproportionately abundant (Supplementary Fig. S1). In addition, zeta decline best fit a power law regression (Fig. 1g), suggesting deterministic, not stochastic factors, primarily drove the composition of these sites at the time of sampling. Variation partitioning analysis suggested community turnover was strongly driven by soil age (28%), soil physicochemistry (5.9%), and the interaction of these factors (18%; Fig. 1h). Overall, these findings suggest both glacier forelands are rapidly and successively colonised by microorganisms through deterministic processes, potentially linked to factors such as nutrient loading, microbial interactions, and soil stabilisation as the glacier retreats.

**Metabolically flexible microorganisms drive glacial foreland colonisation**

We used genome-resolved metagenomics to differentiate the metabolic traits of habitat generalists and specialists in the community. For the Antarctic and Swiss glacier forelands respectively, 367 and 222 species-level (ANI: 0.95) metagenome-assembled genomes (MAGs) were assembled, spanning 17 and 19 phyla and capturing on average 33% and 10% of reads, respectively (Fig. 2a). The specialisation index for each MAG was calculated based on read mapping to the topsoil metagenomes, with MAGs classified as habitat generalists or specialists if their specialisation indices were in the lower or upper quartiles respectively (Supplementary Data 3); this quartile-based classification enabled us to quantitatively discriminate ecologically distinct taxa while ensuring sufficient MAGs were contained within each group to enable robust comparison of their distributions and capabilities. These

MAG-based classifications were generally concordant with the amplicon-based classifications, with clear differentiation of habitat generalists (e.g., Actinobacteriota such as *Gaiella*, Proteobacteria such as *Ramlibacter,* among numerous novel genera) from habitat specialists (e.g., photosynthetic *Microcoleus*, methanotrophic *Methyloglobulus*, Bdellovibrionota and Patescibacteria). However, it also revealed that genus-level classifications sometimes obscured patterns by lumping species with different specialisation indices, for example, in the case of *Polaromonas* with six MAGs spanning the full spectrum of habitat generalists to specialists (Supplementary Data 3). The metabolic versatility of each MAG was inferred based on their total coding sequences and metabolic genes corrected for genome completeness. The habitat generalist MAGs had somewhat larger genomes, encoding on average 4428 and 4267 genes in the Antarctic and Swiss datasets, respectively, compared to the habitat specialist MAGs, at 4026 and 4159 genes (Supplementary Data 4). Similarly, we screened whether they encoded 56 signature metabolic genes (Supplementary Data 4), including the primary dehydrogenases for organotrophy and lithotrophy, the terminal reductases for aerobic and anaerobic respiration, and the signature enzymes for carbon fixation, photosystem- and rhodopsin-based phototrophy, and nitrogen fixation. In line with this trend, generalists encoded more signature metabolic genes than specialists in both ecosystems; however, while this difference was pronounced in the Antarctic dataset (9.6 vs 6.6 per MAG), it was more modest in the Swiss soils (6.8 vs 6.5). This provides the first quantitative support for the hypothesis that metabolic flexibility enhances niche breadth and, in turn, promotes habitat generalism of microorganisms.

As expected from the geochemical analyses of these oxygenated carbon-poor soils (Supplementary Data 1), the MAG-based metabolic annotations suggest both forelands are limited primarily by energy rather than by oxidant and nitrogen supply (Supplementary Data 3). Almost all bacterial MAGs are predicted to mediate respiration using organic compounds as electron donors and oxygen as an electron acceptor, with much of the community also capable of one or more denitrification steps (65% MAGs; adjusted for MAG completion; Supplementary Data 3, 4). There was much capacity for microbes to also input electrons from inorganic and one-carbon compounds from both lithic sources, namely sulfide (19% MAGs), thiosulfate (16%), ammonia (2.3%), nitrite (1.4%), iron (1.6%), as well as the atmospheric trace gases hydrogen (21%), carbon monoxide (10.2%), and methane (0.8%; Supplementary Data 4). Around 8% of MAGs spanning three phyla were predicted to mediate carbon fixation, predominantly using chemosynthetic lineages of RuBisCO, suggesting many are self-sufficient primary producers that use electrons from atmospheric gases and lithic substrates, increasing the organic carbon pool of the system. For instance, at the Antarctic foreland, an abundant proteobacterial MAG in the genus *Polaromonas* co-encoded CO dehydrogenase, thiosulfohydrolase, and reverse dissimilatory sulfite reductase with a form ID RuBisCO, indicating the capacity to use both atmospheric and lithic electron donors to mediate carbon fixation. Despite the minimal

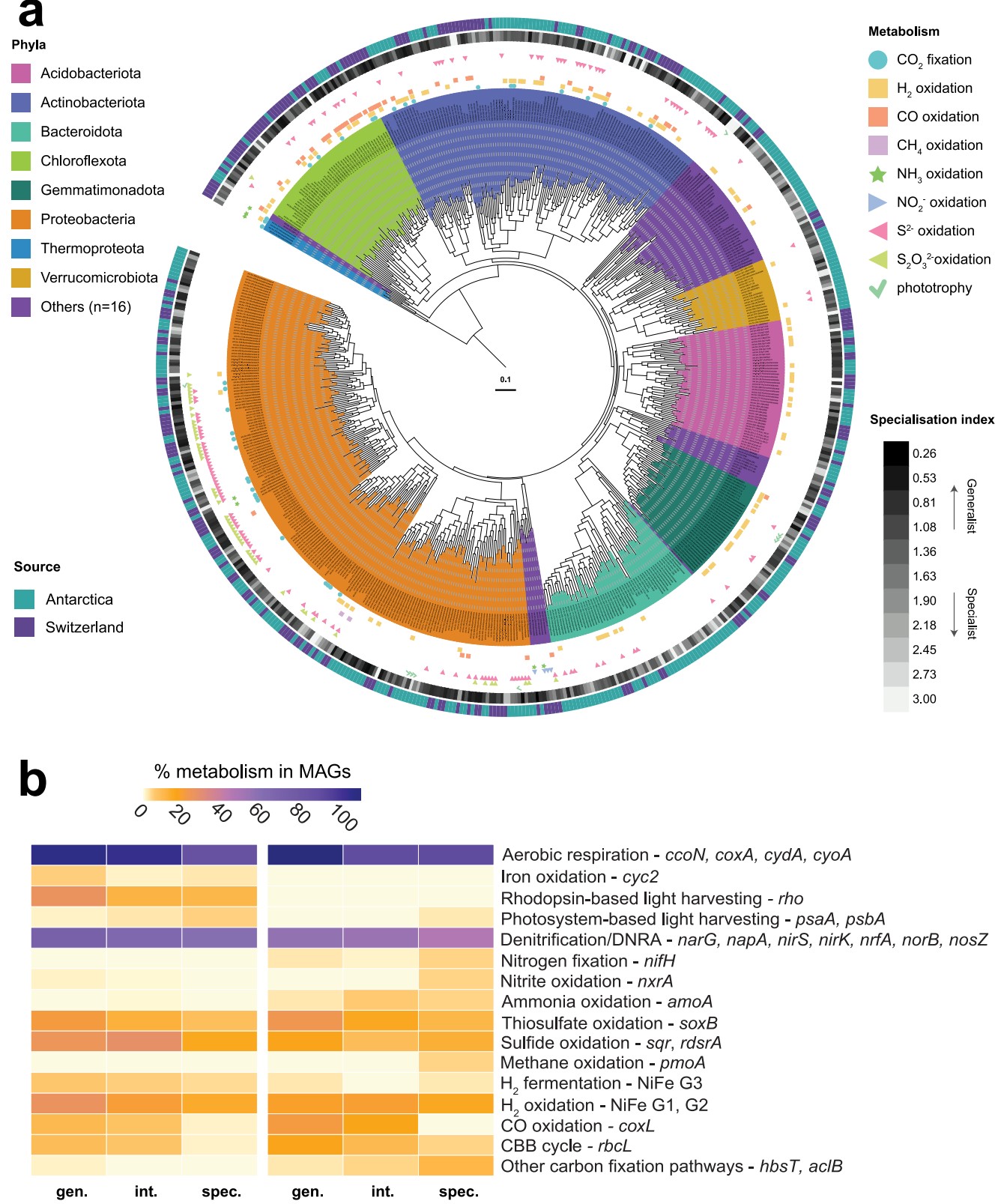

**Fig. 2 | Metabolic traits differentiate habitat generalists and specialists in glacial forelands. a** Phylogenomic tree showing the evolutionary history, metabolic capabilities, and habitat specialisation of the 589 glacial foreland metagenome-assembled genomes (MAGs) spanning 24 phyla. The archaeal and bacterial tree was constructed using the maximum likelihood method, with LG + F + G4 substitution model, with 1000 iterations of ultrafast bootstrapping and midpoint rooting.

Clades are coloured according to phylum-level taxonomy, and symbols indicate the potential of each MAG to mediate key energy and carbon acquisition processes. The heatmap on the inner concentric ring represents the specialisation index, while the outer concentric ring's colours denote the source of each MAG (Swiss vs Antarctic). **b** Metabolic capabilities of MAGs identified as habitat generalists (gen.), intermediates (int.), or specialists (spec.) at the Antarctic and Swiss glaciers.

capacity for oxygenic photosynthesis, 8% of MAGs spanning seven phyla were predicted to grow photoheterotrophically by harvesting light using rhodopsins or photosystem II. Only four MAGs encoded nitrogenases (Supplementary Data 4), suggesting these soils were more strongly carbon- than nitrogen-limiting. Though the metabolic capabilities of the MAGs reconstructed from the two glaciers were similar, an exception was the capacity for phototrophy, which was encoded by 54 Antarctic MAGs (all predicted photoheterotrophs) compared to one Swiss MAG (the cyanobacterium *Microcoleus*; Supplementary Data 4). Conversely, the capacity for nitrification was higher in the Swiss MAGs (Supplementary Data 4). Overall, these data suggest that the primary colonisers of glacier forelands exhibit significant metabolic flexibility, with aerotrophy and classical lithotrophy emerging as dominant processes, likely enabling their adaptation to these oligotrophic settings and providing resources for the establishment of secondary colonisers.

The habitat generalists and specialists differed in their energy and carbon acquisition strategies across the two glaciers (Fig. 2b). For example, RuBisCO genes for carbon fixation through the Calvin-Benson-Bassham cycle were 4.5-fold more abundant in habitat generalists and were almost exclusively associated with chemosynthetic bacteria (Supplementary Data 3, 4). Of these RuBisCO-encoding generalists, 54%, 37%, and 29% were capable of oxidising sulphur compounds (via sulfide-quinone oxidoreductase, reverse dissimilatory sulfite reductases, and thiohydrolase; e.g., *Ramlibacter*), atmospheric $H_2$ (via group 1 h and 1 l [NiFe]-hydrogenases; e.g., Chloroflexota CSP1-4), and atmospheric CO (via form I CO dehydrogenases; e.g., *Pseudonocardia*). The generalist MAGs also had a greatly enhanced capacity for oxidation of inorganic compounds compared to specialists, most strikingly CO (26-fold), but also $H_2$ (1.5-fold), sulfide (1.4-fold), thiosulfate (2.3-fold), and iron (2-fold), as well as rhodopsin-based light harvesting (2.4-fold). This is consistent with the hypothesis that habitat generalists are metabolically flexible, often self-sufficient microorganisms with the capacity to continuously harvest energy and mediate chemosynthesis. Conversely, the genes for photosynthesis, methanotrophy, and nitrification were encoded primarily by metabolically restricted habitat specialists. For example, two seemingly obligately methanotrophic MAGs were retrieved from the Swiss forelands, classified as *Methyloglobulus* and *Methyloligotrophales*, the latter a key aerotrophic clade in desert and cave ecosystems[23]. A notable exception was a Nitrosopiraceae MAG (genus Palsa-1315, also known as Clade B Comammox *Nitrospira*), predicted to be capable of complete nitrification, that was ubiquitous across the Swiss transect and among the 15 most abundant MAGs, and hence can be considered a metabolically restricted pioneer primary producer. Collectively, these findings lend further support to the theory that habitat generalists are usually more metabolically versatile and hence can use a broader range of resources than habitat specialists[21].

## Ecologically and functionally distinct microbes vary in abundance over soil age

We next sought to understand how the ecological traits and metabolic capabilities of the microbial communities varied across the chronosequence. First, we mapped the relative distributions of habitat generalists and specialists with soil age based on both the amplicon and metagenome data. While the absolute abundance of both generalists and specialists increased during succession, their relative proportions shifted. Based on the amplicon datasets, while habitat generalists were abundant across all samples, they formed a lower proportion of the community in the initially exposed soils (av. 49% community) and became dominant in all other soils (av. 70% community) in both forelands (Fig. 3a). This is in line with the observed stabilisation of community structure with time (Fig. 1e). Similarly, for the MAGs data, habitat specialists were most enriched during the initial colonisation (Fig. 3a). Such observations were contrary to our initial

hypothesis, as we expected habitat generalists to drive initial colonisation and habitat specialists to be most dominant in the oldest soils. While a slight increase in the relative abundance of habitat specialists was observed in the oldest soils compared to the medium age soils, their proportion was highest in the youngest soils overall (Fig. 3a). Nevertheless, these observations remain concordant with broader macroecological theory: in line with our recent modification of Grime's theory[42] (CSO: competitor / stress tolerator / opportunist framework), the early-colonising specialists are likely opportunists primed to rapidly colonise new environments by exploiting available resources released during deglaciation, but become outcompeted by stress-tolerant generalists and later-colonising specialists (including competitors) at later successional stages.

These predictions are supported by the analysis of variations of specific taxa with age. Proteobacterial habitat specialists dominate the newly exposed soils, for example, with a *Methylotenera* MAG and a SURF-13 MAG comprising 1.9% and 1.5% of the community (based on metagenomic read mapping), respectively in the earliest Antarctic soils (0 – 5 yrs), but comprise less than 0.1% of the community in more established soils (21 yrs; Supplementary Data 4). Such taxa are also quantitatively more abundant in early rather than late soils (based on 16S rRNA gene copy number normalisation). A similar pattern was observed for the Swiss soils, with MAGs from multiple Burkholderiaceae genera (e.g., *Methylibium*, *Polaromonas*, CADEEN01) abundant in the earliest soils (0 – 7 yrs) but either minimal or undetected in the oldest soils (58–127 yrs; Fig. 3b and Supplementary Data 4). Importantly, these opportunistic primary colonisers are still metabolically flexible, capable of using lithic substrates (e.g., sulfide) and often atmospheric hydrogen to support lithoheterotrophic and sometimes lithoautotrophic growth. The later-colonising habitat specialists, for example, ammonia oxidisers (e.g., Nitrosomonadaceae, Nitrososphaeraceae) and the aforementioned methanotrophs, show the opposite trend and fill metabolic niches unexploited by other colonisers. In contrast, the dominant habitat generalists were also present in the earliest soils, but typically increased modestly in relative abundance and greatly in absolute abundance across each chronosequence; these trends are particularly pronounced for the most abundant lineages of Actinobacteriota (e.g., Nocardioidaceae, Micrococcaceae) and Chloroflexota (e.g., CSP1-4, P2-11E; Fig. 3b), each of which is predicted to be metabolically flexible.

This shift in generalist versus specialist distributions during foreland succession is also reflected by differences in energy and carbon acquisition genes. Based on community-wide profiling using metagenomic short reads (as opposed to MAG-based analysis; Fig. 2), most microbial cells inhabiting the forelands can use inorganic electron donors, including sulfide (av. 44% Swiss topsoil cells, 63% Antarctic cells), $H_2$ (54%, 35%), CO (35%, 28%), and/or ammonium (3.3%, 0.65%) with considerable capacity also for rhodopsin-based light harvesting (4.1%, 21%) and carbon fixation (21%, 19%) (Supplementary Data 3 and Fig. 3c). In line with being encoded primarily by opportunistic early colonisers, the proportion of the community capable of sulfide oxidation strongly and consistently decreased with soil age for both forelands ($p = 8.04 \times 10^{-6}$ Swiss topsoil; $p = 1.06 \times 10^{-10}$ Antarctic), whereas hydrogenases and CO dehydrogenases modestly increased with soil age, in agreement with their association primarily with stress-tolerant habitat generalists (Fig. 3c). Rhodopsin and photosystem genes also decreased with soil age, suggesting that light harvesting can be adaptive for initial colonisation, though this is a less common strategy than lithotrophy. Carbon fixation pathways shifted with soil age in both forelands. The RuBisCO lineages typical of classical chemolithoautotrophs (i.e., form IA, ID and to a lesser extent IC)[64,65] dominate in both forelands in the earliest soil ages (Fig. 3c, d) and, based on genome-resolved analysis, are primarily encoded by sulfide-oxidising Proteobacteria and metabolically flexible Chloroflexota (Supplementary Data 3). In contrast, the form IE RuBisCO typical of

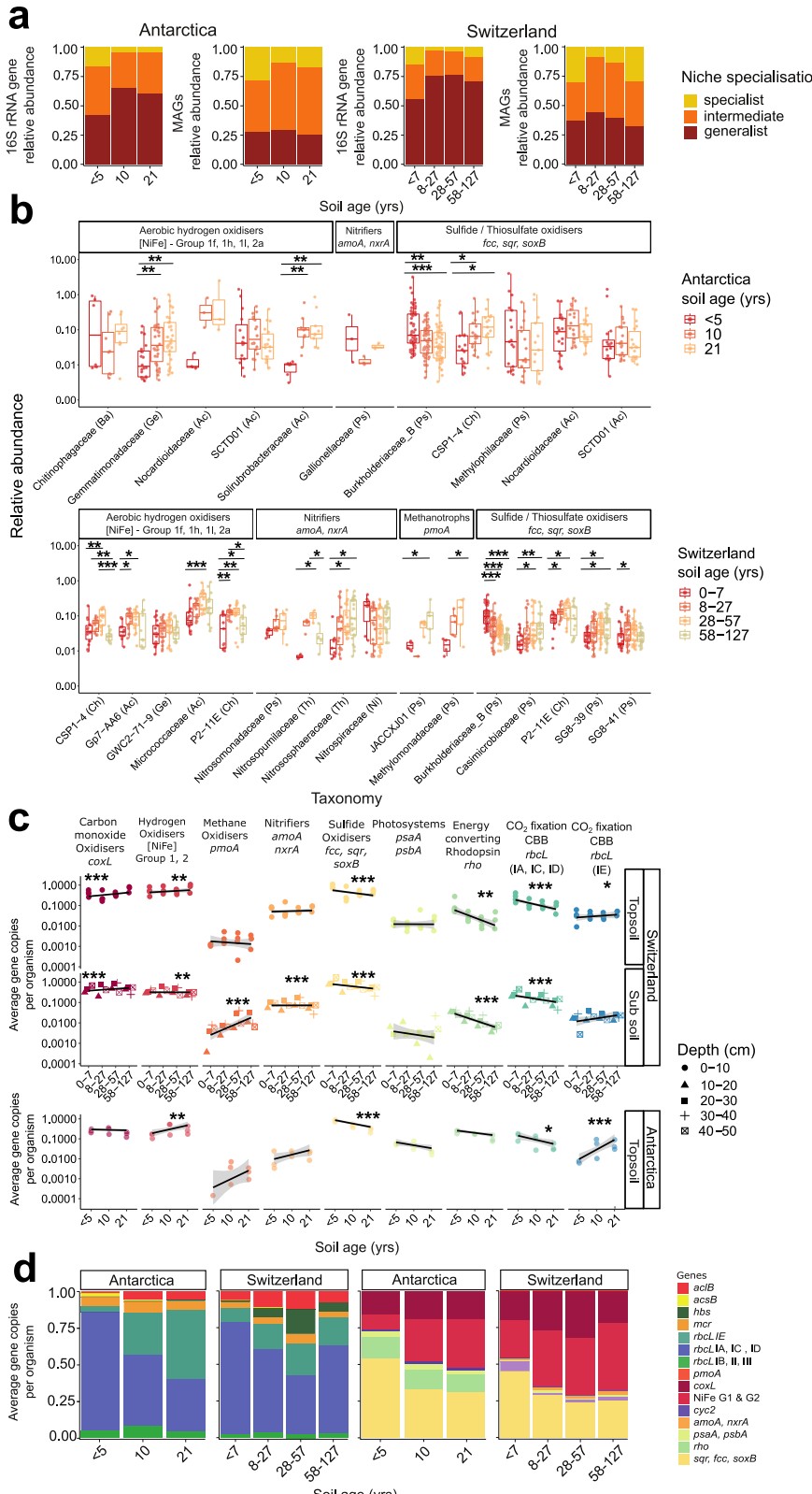

aerotrophs[23,24,66] are encoded exclusively by diverse Actinobacteriota MAGs and become dominant in more established soils (Fig. 3c, d and Supplementary Data 3). Random forest analysis emphasised that soil age since deglaciation is a key driver of the distribution of these genes, especially for sulfide oxidation, in line with a key role of this process in early colonisation of carbon-poor soils, alongside other factors such as pH and nutrient concentrations (Supplementary Fig. S2). Using the Swiss samples, we also examined variations in the levels of metabolic genes with soil depth; while lithic and gaseous energy sources were predicted to be important throughout the soil profiles (Fig. 3c), as expected, phototrophy genes decreased and anaerobic metabolism genes (e.g., for acetogenesis, dehalorespiration, hydrogenogenic fermentation) increased substantially with depth (Fig. 3c and Supplementary Data 3).

**Fig. 3 | Microbial habitat classes and metabolic capabilities vary across the glacial forelands. a** Proportion of specialist, intermediate and generalist microbes classified based on 16S rRNA gene amplicon sequencing and metagenome-assembled-genomes (MAGs) across the Antarctic and Swiss glacier forelands. **b** Boxplots showing the relative abundance of selected MAGs ($n = 114$) predicted to be capable of oxidising lithic and atmospheric substrates across the Antarctic and Swiss glacier forelands. Statistical differences among age groups within MAG families were assessed using a two-sided Kruskal-Wallis test. A Dunn's post-hoc test with Benjamini–Hochberg correction was applied to identify significant pairs, with asterisks denoting significance. Exact $p$-values are provided in the source data. Box plots centre line represents the median, the box bounds denote the 25th and 75th percentiles, and the whiskers extend to the most extreme data points within $1.5 \times$ the interquartile range. **c** Scatter plots showing average gene copy per organism of topsoil (0–10 cm) based on short read analysis of key metabolic marker genes, with a two-sided contrasts generalised linear model for the Swiss and Antarctic glacier sites ($n = 24$). The fitted line represents the linear relationship across age class categories for visualization, with shaded ribbons showing 95% confidence intervals. Asterisks denote significance, and exact $p$-values and model families are shown in Supplementary Data 3. The same is shown for subsoil samples (10–50 cm) of the Swiss glacier ($n = 16$). **d** Proportion of carbon fixation and energy conversion marker genes across the Antarctic and Swiss glacier forelands based on short read analysis.

## Foreland microbial communities oxidise atmospheric and lithic substrates during colonisation

Finally, we conducted biogeochemical analyses to both validate metagenomic predictions and determine energy dynamics of foreland ecosystems during establishment. Based on physicochemical analysis of the soils (Fig. 4a), ammonium, sulphur compounds, and organic carbon are present in sufficient amounts to support lithotrophic and organotrophic growth of high-affinity pioneer microbes; levels of these compounds were low, especially in the Antarctic forelands, except for the sulphur-enriched soils closest to the Swiss alpine glacier, with previous studies suggesting sulphur compounds accumulate due to glacial weathering and meltwater delivery[30,67,68]. Consistent with these observations, both Swiss and Antarctic soils oxidised sulfide in aerobic microcosm assays (Fig. 4b); rates were highest at soils close to the glacier fronts, in line with their enrichment with sulfide-oxidising Proteobacteria (Fig. 3b) and elevated substrate availability at the Swiss but not Antarctic site (Fig. 4a). Reflecting the presence of high-affinity nitrifiers in the glacial forelands, microcosm experiments confirmed that ammonium was oxidised to nitrite and nitrate by the foreland communities under aerobic conditions (Fig. 4c). Ammonium oxidation to nitrate occurred at similar rates throughout the Swiss foreland, in line with the presence of Nitrosopiraceae habitat generalists capable of complete nitrification (Supplementary Data 3). By contrast, nitrite accumulated in the microcosms with the early-stage Antarctic soils but was largely converted into nitrate in the later-stage soils (Fig. 4c), consistent with the metagenomic data that nitrite-oxidising and complete nitrifying bacteria are associated with later stages of succession (Supplementary Data 3). Together, these findings suggest lithic energy sources are used by foreland microbes to support aerobic respiration and likely primary production, with sulfide oxidation potentially serving as a particularly important driver of initial colonisation.

To test our hypothesis that atmospheric trace gases are critical for ecosystem establishment, we also profiled the in situ levels and fluxes of trace gases in the Swiss soils (logistical constraints prevented equivalent analysis in Antarctic soils). H$_2$, CO, and CH$_4$ were present at the soil-atmosphere interface at mixing ratios typical of atmospheric averages ($0.49 \pm 0.04$, $0.37 \pm 0.09$, $1.8 \pm 0.1$ ppmv respectively) and at variable levels at 1 m depth ($0.074 \pm 0.040$, $0.50 \pm 0.32$, $0.56 \pm 0.43$ respectively); multiple factors influence the in situ levels of these gases, including gas exchange and diffusion rates in soil, biological consumption, and abiotic and biotic production processes, though the sharp depth-related decreases for H$_2$ and CH$_4$ suggest consumption by hydrogenotrophs and methanotrophs (Fig. 4d). Consistently, in situ flux profiling revealed both gases were consumed, albeit at differential rates across the foreland. H$_2$ was consumed at rapid rates across the whole foreland chronosequence with an average threefold increase from early to late deglaciated soils (Fig. 4e), in line with the gene- and genome-centric data. In contrast, uptake of CH$_4$ established at later successional stages, increasing on average 17-fold from early to late deglaciated soil (Fig. 4e), supporting previous measurements[69] and the observed distribution of methanotrophs and their marker genes (Figs. 1–3). As with our previous investigations of organic soils[19], we did

not observe clear trends in CO levels or fluxes (Fig. 4e) despite the widespread distribution of CO dehydrogenase (Fig. 3c) and high ex situ CO oxidation activities (Fig. 4f), with abiotic production within the flux chamber potentially obscuring trends. These findings support that trace gases are critical energy sources in these oligotrophic environments, with H$_2$ oxidation emerging as an early-establishing, generalist-associated trait and CH$_4$ oxidation as a later-establishing, specialist-associated one.

To help resolve to what extent atmospheric trace gases support foreland development, we measured their oxidation in Swiss and Antarctic samples using ex situ microcosms. Overall, the Swiss and Antarctic soil showed different trends in trace gas consumption (Fig. 4f, g), with oxidation rates at the Swiss glacier generally declining with soil depth (Supplementary Fig. S3). For the Swiss samples, CH$_4$ oxidation greatly increased with soil age, while H$_2$ and CO oxidation occurred at consistent rates across the chronosequence (Fig. 4f). In contrast, oxidation rates increased with soil age for the Antarctic soil, in part driven by an increased absolute abundance of H$_2$ and CO oxidisers during succession (Fig. 4g). Although previous studies have shown forelands can be a sink for methane[70], this is the first evidence of H$_2$ and CO uptake in these communities. Building on these results, we investigated whether microbial communities could rely on H$_2$, CO and CH$_4$ for their energy demands (Supplementary Fig. S3). To do so, we used thermodynamic modelling to calculate the amount of power (J s$^{-1}$ (W)) per cell that could be generated based on bulk trace gas oxidation rates (Supplementary Data 5) and the total number of putative trace gas oxidizers detected per gram of dry soil (Supplementary Data 2 & 3). While rates varied between gases and sites, the average power per cell ($3.83 \times 10^{-14}$ W cell$^{-1}$; range $3.16 \times 10^{-19}$ to $2.11 \times 10^{-12}$ W cell$^{-1}$) is within the range to support the maintenance of all cells in the community (noting empirical measurements of maintenance energy range from $10^{-12}$ to $10^{-17}$ W cell$^{-1}$ based on previous studies, with levels much lower in oligotrophs[71–74]). For some cells, the power derived from trace gas oxidation is sufficient to support mixotrophic or aerotrophic growth[75]. Altogether, these findings suggest that trace gases have a major role both in supporting colonisation and subsequent development of a new terrestrial ecosystem.

## Discussion

Despite their vastly different geographical locations and physico-chemical properties, the primary colonisers of the Antarctic maritime and Swiss alpine glacier foreland were remarkably similar in functional capabilities. Rather than photosynthetic microbes, the most abundant and active ecosystem engineers appear to be versatile chemosynthetic bacteria that use atmospheric and lithic substrates. These findings highlight the central role of metabolic flexibility in shaping colonisation and succession of new ecosystems. On one hand, metabolic flexibility provides opportunistic specialists with the capacity to rapidly exploit the varied resources liberated as glaciers retreat. On the other hand, this flexibility also provides habitat generalists with the capacity to adapt to the extensive physical, chemical, and biological changes that occur during different stages of ecosystem development. In both glaciers, there has been extensive selection for microbes that

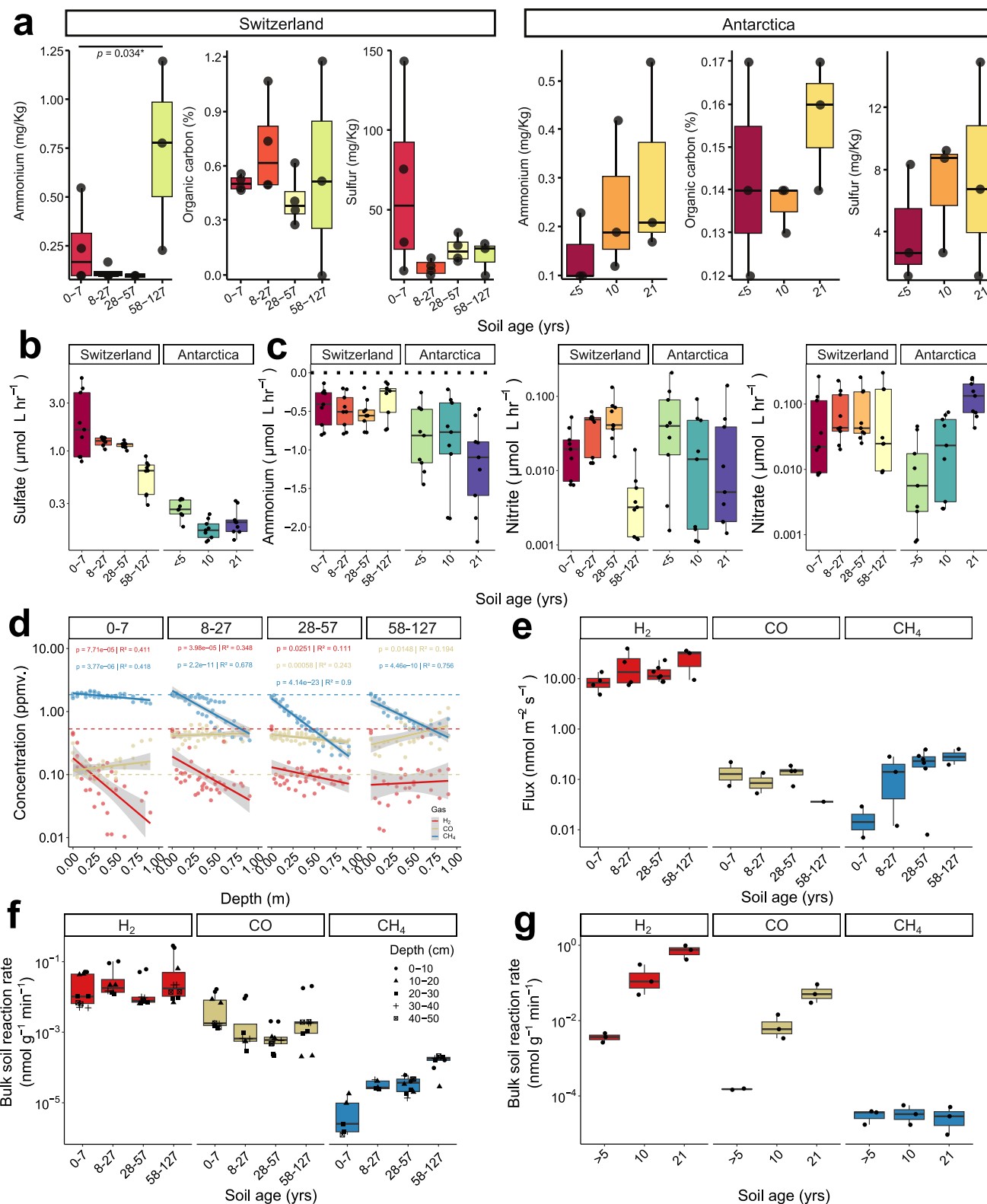

simultaneously or alternatively use organic, lithic, and atmospheric electron donors. Similar dynamics appear to influence the quantitative ecology, metabolic capabilities, and biogeochemical activities of the microbes inhabiting both forelands. Key differences include a more pronounced increase in microbial abundance and diversity over the Antarctic chronosequence, as well as potentially a stronger role for phototrophy in the Antarctic soils and ammonia- and sulfide-based lithotrophy in the Swiss soils. This convergence of findings suggests

that microbial community assembly in glacier forelands is not primarily stochastic, but rather deterministic, as also supported by ecological and process modelling. The dynamic interplay between physicochemical conditions and biological interactions along the glacier chronosequences causes environmental filtering of communities, shifting specialist-generalist dynamics to enable co-existence of metabolically versatile habitat generalists with more restricted habitat specialists. Consistently, we observed increasing compositional

**Fig. 4 | Trace gases and lithic compounds are major substrates that are differentially consumed across the chronosequences. a** Boxplots show nutrient concentrations across the Swiss ($n = 15$) and Antarctic ($n = 9$) glacier forelands, with measurements taken from three biological replicates per age group. Statistical differences among age groups for each nutrient were determined using two-sided ANOVA or the Kruskal-Wallis test. **b, c** Ex situ oxidation rates of sulphur (**b**) and nitrogen (**c**) species were measured in microcosm experiments containing 10 g of soil and 200 mL of ultrapure water, supplemented with 50 μM of ammonium chloride or sodium sulfide. Positive values indicate accumulation, while negative values indicate uptake. Measurements were performed with at least three biological replicates per age group. **d** Scatter plots illustrate in situ soil gas concentrations at Swiss sites relative to soil depth spanning ambient atmospheric conditions

to deep subsoils (0–1 m). Linear models depicting significant relationships annotated with exact $p$-values and $R^2$ coefficients, with shaded ribbons showing 95% confidence intervals. **e** In situ soil–atmosphere gas fluxes of Swiss samples for each gas, with a minimum of three biological replicates per age group. **f** Bulk soil oxidation rates for each gas by soil age in Swiss samples, with different shapes indicating respective soil depths. Measurements were conducted on a minimum of five biological replicates per age group. **g** Bulk soil oxidation rates for each gas by soil age in Antarctic samples, measured with three biological replicates per age group. Box plots centre line represents the median, the box bounds denote the 25th and 75th percentiles, and the whiskers extend to the most extreme data points within $1.5 \times$ the interquartile range.

stability over time, as also observed for other established glacier foreland microbial communities[53,76–78].

In addition to deepening knowledge of the ecological dynamics of primary succession, by applying macroecological theory to microbial systems, this study also uncovers that atmospheric trace gas oxidation is a critical yet previously overlooked means for microorganisms to establish new ecosystems. This was confirmed through extensive metagenomic, flux, and activity measurements, as well as theoretical modelling. We also provide the first quantitative evidence that microbial metabolic flexibility is correlated with habitat generalism. It should be noted that, whereas habitat generalism is a continuous distribution, this study primarily focused on the upper and lower quartiles of this distribution; future studies should explore the distributions and capabilities of microbes across the full spectrum of generalism by applying continuous analyses or finer percentile-based classifications. In addition, while we observed similar findings across two divergent glacial forelands, it is unclear to what extent these findings predict colonisation in other forelands, as well as primary and secondary succession in other ecosystems (e.g., volcanic soils, meteorites, post-fire recovery). Future work is also needed to disentangle the sources of microbes and the interplay of dispersal with selection during initial colonisation. Nevertheless, these findings are timely considering the current unprecedented rates of glacier decline worldwide and the expansion of forelands. The microbial dynamics during the earliest stages of colonisation and succession are critical to the eventual fate of these environments, as they develop into dynamic ecosystems harbouring biodiversity and driving globally-relevant biogeochemical cycles.

# Methods
## Field sampling
Soil samples were collected from the forelands of two retreating glaciers: Sally Rocks tongue of Hurd Glacier on Livingston Island in Antarctica (March 2022) and Griessfirn Glacier in Canton Uri, Switzerland (August 2022). The Hurd Glacier is located at 62.6996°S, 60.4156°W, while the Griessfirn Glacier is at 46.84299°N, 8.82746°E. Specific coordinates for each sampling site at both glaciers are provided in Supplementary Data 1. Sampling permits for the Griessfirn glacier forefield were not required, whereas Antarctic sample collection was conducted under permit CPE-2020-9, issued by the Spanish Antarctic Committee, in accordance with Annex II of the Protocol on Environmental Protection to the Antarctic Treaty. Previous studies at both sites were used to inform the sampling design, including information on how deglaciation age was determined[31,50,79]. For the Antarctic site, nine surface soil samples (0-10 cm) were collected, corresponding to three soil age classes determined based on glacier front position measurements[80,81]: A (< 5 yrs), B (10 yrs), and C (21 yrs). Samples for 16S rRNA gene and metagenomic analyses were collected with a sterile spade, placed in whirl-pak sampling bags, frozen, and shipped to Monash University for processing. For Swiss samples, sampling locations were distributed along a well-defined soil chronosequence with increasing distance from the glacier terminus and categorised into

four soil age classes: A (0–7 yr), B (8–27 yr), C (28–57 yr) and D (58–127 yr). In addition, at one location in each of the four soil-age classes, a 50 cm soil profile was excavated to allow depth-resolved soil sampling at 10 cm intervals. Sampling locations were positioned along SW-NE-oriented transects that followed a single geomorphological landform, a band of lateral debris deposits, to minimise variability due to allogenic factors and intrinsic differences in microclimatic conditions and physical parameters among landform types[69]. Sampling was performed under dry-weather conditions and distant from previous rainfall. All samples were kept on ice or refrigerated until arrival in the laboratory and were thereafter stored at − 20 °C (for molecular analysis) or 4 °C (for ex situ incubations) until further processing.

## Soil physicochemical analysis
Surface soils (0–10 cm) from of the Swiss and Antarctic sites were analysed at the Environmental Analysis Laboratory (EAL), Southern Cross University. In total, 34 soil physicochemical parameters were selected for analysis, based on commonly reported drivers of soil microbial composition globally. These included: phosphorus (mg/kg P), nitrate nitrogen (mg/kg N), ammonium nitrogen (mg/kg N), sulphur (mg/kg S), pH, electrical conductivity (dS/m), estimated organic matter (% OM), exchangeable calcium, exchangeable magnesium, exchangeable potassium, exchangeable sodium, exchangeable aluminium, exchangeable hydrogen, effective cation exchange capacity (ECEC) (cmol + /kg), calcium (%), magnesium (%), potassium (%), sodium - ESP (%), aluminium (%), hydrogen (%), calcium/magnesium ratio, zinc (mg/kg), manganese (mg/kg), iron (mg/kg), copper (mg/kg), boron (mg/kg), silicon (mg/kg Si), total carbon (%), total nitrogen (%), carbon/nitrogen ratio, basic texture, basic colour, chloride estimate (equiv. mg/kg), and total organic carbon (%).

## In situ gas and flux measurements
For the Swiss glacial chronosequence, in situ depth-resolved soil-gas samples were collected at 11 locations using a poly-use multilevel sampling system[70]. Details of its installation, sampling, and gas measurement procedures were described previously[82]. Concurrent with soil-gas sampling, we measured the depth-resolved volumetric soil-water content (cubic metres per cubic metre of soil; PR2/6 capacitance probe; Delta-T Devices Ltd., Cambridge, UK) and recorded depth-resolved soil temperature (iButton temperature loggers; Maxim Integrated, San Jose, CA, USA) in 30-minute intervals throughout the duration of the sampling campaign. At 18 locations, we used static flux chambers to quantify trace gas fluxes. Details of the installed chambers, sampling and measurement procedures were described previously[19,69]. In this study, the deployment of the collars at the new locations was performed prior to the start of the sampling campaign and included rainfall events to allow soil consolidation around the collars. Each flux chamber measurement lasted 90 min and consisted of nine gas samples collected at the following time intervals: four samples were collected within the first 10 min at nearly regular intervals, followed by five samples collected at increasingly longer intervals (10, 15, and 30 min). All gas samples were kept cooled until the

measurement of trace gas concentrations by gas chromatography using a pulsed discharge helium-ionisation detector (model TGA-6791-W-4U-2, Valco Instruments Company Inc.) as previously described[19]. Once the measurements were completed, topsoil samples were collected with a sterile spade at the flux-chamber locations for DNA extraction, quantitative polymerase chain reaction (qPCR) and metagenomic sequencing, and for ex situ incubations to quantify trace-gas oxidation rates using a previously described sampling procedure[19,70]. Finally, soil bulk-density measurements were performed at eight locations applying an adapted PU-foam method suitable for soils with high skeletal fraction[83] as previously described[50].

### Ex situ oxidation rates
To assess the capacity of the microbial communities at each glacier to oxidise trace gases ($H_2$, CO, $CH_4$), we placed the surface soils from the Swiss and Antarctic chronosequences and the soils from the Swiss depth profile samples in 120 ml serum vials. The wet weight of soil in each microcosm was recorded and used to normalise subsequent calculations. The headspace was continuously purged with air from a pressurised cylinder (Air Liquide) to achieve mixing ratios that reflect atmospheric levels, specifically 0.5 ppm $H_2$, 0.6 ppm CO, and 1.8 ppm $CH_4$. Sampling commenced immediately after sealing the vial, and headspace samples of 2 mL were taken at five intervals. Heat-killed soils (subject to two autoclave cycles at 121 °C for 30 min each) and blank measurements using empty serum vials served as controls. These controls confirmed that trace gas oxidation in the samples was attributable to biotic processes. Gas concentrations were measured using the aforementioned gas chromatograph as previously described[19]. The power per cell derived from the oxidation of each trace gas was calculated as previously described[19,25], by factoring in the reaction rate for each gas (based on ex situ oxidation rates at atmospheric concentrations), Gibbs energy of the reaction (based on standard Gibbs energy, the reaction quotient, gas-phase compound activities, and soil conditions), and the number of microbial cells involved (based on 16S rRNA gene copies quantified by qPCR and proportion of trace gas oxidising cells determined by metagenomics).

### Ammonium and sulfide oxidation
Oxic slurry experiments were undertaken to determine the oxidation rates of ammonium and sulfide in surface soils collected from both glaciers. Sterilised surface soils (autoclaved at 120 °C for 1 hour) were used as controls to confirm that the observed oxidation rates were driven solely by biotic processes. Slurries containing 10 g soil (wet weight) and 200 mL substrate amended-ultrapure water (50 μM of either ammonium chloride or sodium sulfide) were prepared in 250 mL Schott bottles. The slurries were aerated for 5 min to ensure oxic conditions. The bottles containing the slurries were left uncapped but loosely covered with pre-combusted aluminium foil. The slurries were incubated in the dark and were mixed periodically for the duration of the incubation period (up to 22 days). At each time point, 15 mL of samples were collected and filtered through 0.22 μm pore-sized filters (Sartorius Minisart syringe filter). Filtered samples were analysed for ammonium, nitrate, nitrite and sulfate concentrations using a Lachat Quickchem 8000 Flow Injection Analyser following the procedures in Standard Methods for Water and Wastewater (APHA 2005)[84]. Rates for the oxidation of ammonium and accumulation of nitrite, nitrate and sulfate over time were calculated using linear and non-linear regression. Best model fits were determined using AIC to select the final rates.

### Chlorophyll *a* measurement
To extract chlorophyll from each sample, 9 mL of acetone was added to 5 g of soil, mixed, sonicated in an ultrasonic water bath at a maximum setting for 5 min, and incubated overnight in a refrigerator. Then, 1 mL of RO water was added to each sample, and these were

subsequently centrifuged for 10 min at 447 x g. 3 mL of the liquid phase of the sample extract were transferred to a cuvette, and chlorophyll was measured on a spectrophotometer (Eppendorf BioSpectrometer) with wavelength scan at 665 and 750 nm.

### Community DNA extraction
Total community DNA was extracted for each of the Swiss and Antarctic samples using 0.5 g of soil. Extractions were performed using the MoBio PowerSoil Isolation kit according to the manufacturer's instructions, with samples eluted in DNase- and RNase-free UltraPure Water (ThermoFisher). A sample-free negative control was also run for each glacier sampled. Nucleic acid purity, yield, and integrity were measured using a NanoDrop ND-1000 spectrophotometer, a Qubit Fluorometer 2.0, and through agarose gel electrophoresis.

### Quantitative PCR
Quantitative polymerase chain reactions (qPCR) were used to estimate total bacterial and archaeal biomass of biocrust and topsoil samples. The 16S rRNA gene was amplified using the degenerate primer pair (515 F 5′-154 GTGY- CAGCMGCCGCGGTAA-3′) and 806 R 5′-GGAC-TACNVGGGTWTCTAAT-3′). A synthetic *E. coli* 16S rRNA gene sequence in a pUC-like cloning vector (pMA plasmid; GeneArt, ThermoFisher Scientific) was used as a standard. PCR reactions were set up in each well of a 96-well plate using LightCycler 480 SYBR Green I Master Mix. Each sample was run in triplicate and standards in duplicate on a LightCycler 480 Instrument II (Roche). The qPCR conditions were as follows: pre-incubation at 95 °C for 3 min and 45 cycles of denaturation 95 °C for 30 s, annealing at 54 °C for 30 s, and extension at 72 °C for 24 s. 16S rRNA gene copy numbers were calculated based on a standard curve constructed by plotting average $C_p$ values of a serial dilution of the plasmid-borne standard against their copy numbers.

### Community profiling
16S rRNA gene amplicon sequencing was used to infer the composition of the bacterial and archaeal community in each sample. Specifically, the V4 hypervariable region of the 16S rRNA gene was amplified using the universal primer pairs F515 and R806 from the Earth Microbiome Project[85]. Illumina paired-end sequencing (2 × 300 bp) was conducted at the Australian Centre for Ecogenomics, University of Queensland. The resulting raw sequences underwent quality filtering, primer trimming, denoising, and removal of singletons within the QIIME 2 platform[86]. The final dataset comprised 14,271 high-quality 16S rRNA amplicon sequence variants (ASVs).

### Biodiversity analyses
Alpha and beta diversity were calculated using R v 4.1.0 and the Phyloseq package[87]. First, reads were normalised using the coverage-based rarefaction and extrapolation method implemented in the R package iNEXT[88]. Coverage was calculated for each sample using the function phyloseq_coverage, followed by rarefaction of all reads, using the function phyloseq_coverage_raref with default parameters. Estimated richness was calculated using the estimate_richness function specifying "Shannon" flag. To calculate beta diversity, sample counts were first transformed to relative abundance and ordinated using a Principal Coordinates Analysis with the ordinate function and the "PCoA" and "bray" options. To test for significant difference in microbial community structure between soil age and location a Permutational Multivariate Analysis of Variance Using Distance Matrices was used via the adonis function using the R package *vegan*[89]. Finally, to determine if significant results were because of dispersion a Multivariate homogeneity of groups dispersions test was used via the *betadisper* function. The multisite metric zeta diversity (ζ) was used to assess incidence-based turnover in community composition (ASVs) along the Swiss glacier foreland using the *zetadiv* R package[62]. $\zeta_2$ and $\zeta_4$ distance decay were computed via the *zeta.ddecay* function on a Jaccard-normalised

presence-absence ASVs dataset with subsampling set to 1,000. Zeta decline was calculated with the function *zeta.decline.mc*, performing subsampling set to 1,000 on a Jaccard-normalised presence-absence ASVs dataset for ζ orders $\zeta_1$ to $\zeta_6$. This approach captured community structuring across the foreland, with increasing ζ values approaching zero. Power-law and exponential models were fitted to the ζ decline curves, using Akaike Information Criterion (AIC) scores to evaluate which model best explained the relationship between ζ diversity and order *i*. Zeta variation partitioning was performed via the function *zeta.varpart* on a presence-absence ASVs dataset using *method.glm = glm.fit2*. To determine the optimal model for *zeta.varpart*, we used redundancy analysis computed via the *rda* function in the *vegan*[89] R package, retaining only non-collinear physicochemical variables.

## Metagenomic sequencing and metabolic profiling

For the Swiss samples, metagenomic shotgun libraries were prepared using the Nextera XT DNA Sample Preparation Kit (Illumina Inc., San Diego, CA, USA) and subject to paired-end sequencing (2 × 150 bp) on an Illumina NovaSeq6000 platform at the Australian Centre for Ecogenomics (ACE), University of Queensland. For the Antarctic samples, metagenomic shotgun libraries were prepared and subject to paired-end (2 × 100 bp) on a DNBSEQ-G400 platform at Micromon Genomics, Monash University.

## Metagenomic short read analysis

To estimate the metabolic capability of the soil communities, quality-filtered short reads were searched against custom protein databases of 56 metabolic marker genes (https://bridges.monash.edu/collections/Greening_lab_metabolic_marker_gene_databases/5230745). These databases contain the amino acid sequences of an essential (usually catalytic) subunit of a key enzyme for each of the major pathways of energy, carbon, and nitrogen acquisition. They include enzymes catalysing the six pathways of carbon fixation, photosystem- and rhodopsin-based phototrophy, aerobic and anaerobic respiration, and nitrogen, sulphur, iron, hydrogen, carbon monoxide, and methane cycling, as listed in Supplementary Data 3. As previously described, these databases were carefully manually constructed to encompass the known diversity of each enzyme[90–92]. DIAMOND v.0.9.31 searches[93] were conducted with a query coverage of 80% and identity threshold of 50%, except for *rho* (40%), *nuoF*, group 4 [NiFe]-hydrogenases, *mmoX*, [FeFe]-hydrogenases, *coxL*, *amoA*, *nxrA*, *rbcL* (all 60%), *psaA* (80%), *psbA*, *isoA*, *atpA*, *aro*, *ygfK* (70%), *and hbsT* (75%). These carefully curated databases and search criteria enable metagenomic short reads encoding each metabolic gene to be accurately quantified. The average gene copy per organism is calculated by dividing the read counts for each gene (in reads per kilobase million, RPKM) against the read count for universal single-copy ribosomal protein genes in RPKM, as previously described[19,25,94].

## Metabolic gene driver analyses

Generalised linear models (GLMs) were used to determine the effects of soil age and depth on metabolic gene abundance based on metagenomic short reads. Following mean–variance comparisons to detect overdispersion and comparing optimal model fits against the residual plots of each distribution. Other important environmental predictors were assessed using a random Forest analysis via the R package randomForest[95]. Collinearity between predictor variables was assessed by constructing Pearsons's correlation matrices using the *cor* function in base R, between all soil physicochemical components measured in this study. Variable reduction was conducted through an ecologically informed iterative process. Initially, correlations among potential predictors were calculated. Highly correlated pairs ($r \geq 0.9$) were manually inspected. Following each correlation analysis, the less ecologically significant variable of a highly correlated pair was removed, based on established knowledge of key environmental drivers of microbial communities in aerated soil ecosystems[96]. For instance, if zinc and pH were found to be highly correlated, pH would be retained due to its recognised overriding ecological importance. This process was repeated to refine the set of variables, resulting in a final set of predictors which were used for subsequent analyses. A random forest model was then generated, assessing the importance of each predictor against trace gas oxidation rates, using the *randomForest* function. Number of variables randomly sampled as candidates at each split = 10, number of trees grown = 10,000 and sampling of cases was conducted with replacement.

## Metagenome-assembled genome analysis

Using the Metaphor pipeline[97], the metagenomes of the Antarctic and Swiss datasets were co-assembled independently of each other using MEGAHIT v1.2.9[98] with default settings. Contigs shorter than 1,000 bp were discarded. The assembled contigs were binned with Vamb v4.1.3[99], MetaBAT v2.12.1[100], CONCOCT v1.1.0[101] and SemiBin2[102]. The four bin sets were subsequently refined with MetaWRAP's refinement module[103] and de-replicated with dRep v3.4.2[104] with 95% ANI integrated with CheckM2[105]. The completeness and contamination of the bins were assessed using CheckM2[105]. Quality thresholds for bins were selected based on a previous study[106], retaining only medium (completeness > 50%, contamination < 10%) and high (completeness > 90%, contamination < 5%) quality bins for further processing, which were termed metagenome-assembled genomes (MAGs). MAGs taxonomy was determined using the Genome Taxonomy Database Release R214[107] via GTDB-Tk v2.3.2[108]. Proteins from each MAG were predicted intrinsically in CheckM2[105]. The MAGs were metabolically annotated against the above-described custom database using DIAMOND v.0.9.31. Gene hits were filtered to retain only those either at least 40 amino acids in length or with at least 80% query or 80% subject coverage. For predicted proteins, the same identity thresholds were used as above except for *atpA* (60%), *psbA* (60%), *rdhA* (45%), *cyc2* (35%), and *rho* (30%).

## Phylogenetic analysis

The genome phylogenetic tree, comprising high- and medium-quality MAGs (completion > 50% and contamination < 10%) bacterial and archaeal MAGs, was constructed using PhyloPhlAn v3.0.67[109]. Multiple sequence alignment was performed using MAFFT v7.508[110] by identifying 400 universal proteins from these microbial genomes, following Segata et al. [111]. The tree was inferred with IQ-TREE v. 2.2.0.3[112] using model LG + F + G4 for and 1,000 ultrafast bootstrap iterations. The tree was midpoint rooted and rendered in iTOL v7[113]. A total of 589 MAGs were analysed using PhyloPhlAn v3.0.67[109], of which 574 were retained after multiple-sequence alignment. The remaining MAGs were excluded during alignment against PhyloPhlAn's internal marker-gene database, likely due to incomplete marker representation. The majority of the filtered MAGs belonged to Patescibacteria, a lineage characterised by small, gene-poor genomes, or exhibited low estimated completeness.

## Habitat specialisation indices

The degree of habitat specialisation of each MAG (i.e., whether they were relative 'habitat generalists' or 'habitat specialists') was calculated based on the coefficient of variance in the metagenomes (i.e., relative mean divided by relative standard deviation calculated by read mapping in CoverM v0.7.0[114]). This metric, termed the 'habitat specialisation index', was originally proposed in macroecology[61,115] and has since been effectively applied in microbial systems[21]; it quantifies niche breadth in a way that is simple, continuous, and extendible across data types (e.g., amplicon and metagenomic datasets), supporting a macroecological framework in microbial ecology. The community-wide mean and standard deviation specialisation index was $1.459 \pm 0.621$ for the Antarctic MAGs and $1.302 \pm 0.792$ for the Swiss MAGs. Relative

'habitat generalists' were defined as MAGs with a specialisation index (i.e., coefficient of variance) below the first quartile (below 0.9886 for Antarctic MAGs, 0.6652 for Swiss MAGs), relative 'habitat specialists' were defined as MAGs with a specialisation index in the fourth quartile (above 1.8751 for Antarctic MAGs, 1.7778 for Swiss MAGs), and relative 'habitat intermediates' are within the interquartile range. Inspired by previous studies in microbial ecology[116], these quartile-based thresholds enable discrimination of ecologically distinct groups within a continuous distribution, while ensuring sufficient taxa are contained within each group for meaningful statistical comparisons. In contrast, more extreme thresholds would reduce statistical power, while broader cutoffs would dilute the ecological signal. To disentangle the effects of soil age and depth for the Swiss samples, the habitat specialisation index was only calculated based on read mapping to the surface metagenomes. This quartile-based specialisation index was also used to classify taxa based on the 16S rRNA gene amplicon sequencing data. Specifically, the habitat specialisation indices of each genus and phylum with an average relative abundance exceeding 0.005% was also calculated, based on their coefficient of variance; again, habitat generalists and specialists defined as taxa with specialisation indices below the first quartile and above the fourth quartile, respectively.

## Reporting summary

Further information on research design is available in the Nature Portfolio Reporting Summary linked to this article.

## Data availability

All sequence data from this study is available at the Sequence Read Archive, with accession numbers PRJNA1178459 for metagenomic sequences, PRJNA1178814 for 16S rRNA gene amplicon sequences, PRJNA1206715 for Antarctic MAGs, and PRJNA1206714 for Swiss MAGs. Source data are provided as a Source Data file. Codes can be shared on request. Source data are provided in this paper.

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

## Acknowledgements

This study was supported by an NHMRC EL2 Fellowship (APP1178715; to C.G.), the Human Frontiers Science Programme (RGY0058/2022; to C.G. and J.A.B.), ARC Discovery Early Career Research Award (DE230101346; to S.K.B., DE250101210; to P.M.L.), a Monash University Early Career Postdoctoral Fellowship (to F.R.), an Australian Research Council SRIEAS Grant SR200100005 Securing Antarctica's Environmental Future (to C.G.), a Swiss National Science Foundation Early Mobility Postdoctoral Fellowship (to E.C.), NERC (NE/T010967/1), the Agence Nationale de la Recherche (ANR23-CPJ1-0172-01) and the European Research Council (ERC) under the European Union's Horizon Europe Research and Innovation programme (Grant agreement No. 101115755) (to J.A.B.), and a Agencia Estatal de Investigación grant (PID2019-105469RB-C22 and PID2023-147027NB-I00B; to A.d.l.R.). We thank Steven L. Chown for helpful discussions.

## Author contributions

E.C., C.G., P.A.N. and M.H.S. conceived and designed this study. C.G., M.H.S. and P.L.M.C. supervised this study. E.C., P.A.N., M.H.S., B.F.M. and A.d.l.R. conducted fieldwork. E.C., S.K.B., P.A.N., W.W.W., L.J., T.J., G.N., V.E., and M.H. conducted and analysed biogeochemical assays. S.K.B., F.R., C.G., and E.C. conducted community and biodiversity analyses. F.R., C.G., G.N., S.K.B., and P.M.L. conducted metagenomic analyses. J.A.B and A.S. provided ecological insights. F.R. and C.G. wrote the paper with input from all authors.

## Competing interests

The authors declare no competing interests.

## Additional information

¹Department of Microbiology, Biomedicine Discovery Institute, Monash University, Melbourne, VIC, Australia. ²Securing Antarctica's Environmental Future, Monash University, Melbourne, VIC, Australia. ³Department Microbiology, Anatomy, Physiology & Pharmacology, La Trobe University, Melbourne, VIC, Australia. ⁴Water Studies Centre, School of Chemistry, Monash University, Melbourne, VIC, Australia. ⁵Aix Marseille Université, Université de Toulon, CNRS, IRD, MIO, Marseille, France. ⁶School of Biological and Behavioural Sciences, Queen Mary University of London, London, UK. ⁷Department of Plant Biology and Ecology, University of the Basque Country (UPV/EHU), Leioa, Spain. ⁸Museo Nacional de Ciencias Naturales, Consejo Superior de Investigaciones Científicas, Madrid, Spain. ⁹Institute of Biogeochemistry and Pollutant Dynamics, ETH Zürich, Zürich, Switzerland. ¹⁰School of Biological Sciences, Monash University, Melbourne, VIC, Australia. ¹¹These authors contributed equally: Francesco Ricci, Sean K. Bay. ✉e-mail: eleonora.chiri@agromini.org; chris.greening@monash.edu

