## [Transparent Peer Review file · Nature Communications]

Metabolically flexible microorganisms rapidly establish glacial foreland ecosystems

Corresponding Author: Professor Chris Greening

Version 0:

Reviewer comments:

Reviewer #1

(Remarks to the Author)

The draft article by Ricci et al. provides excellent points that have implications for how ecosystems are formed. The authors investigated the functional basis of microbial colonization in the Antarctic and an alpine Swiss retreating glacier by integrating data from genome resolved metagenomics, quantitative ecology, and biogeochemical data. The paper provides interesting data and concepts for how generalists and specialist microbial communities colonize during succession. The paper is of high importance and relevance to this journal. It is also exceptionally well written and communicated.

The major concern with the manuscript is regarding the different indices to determine habitat specializations (Relative Habitat Use, RHU). The authors used the coefficient of variance to determine generalist and specialist species, using both amplicon and metagenomics sequences. We strongly recommend the authors provide a more concrete basis and justification, in addition to more relevant citations why they used the first quartile and fourth quartile as threshold to determine species specialization, it would be helpful for the manuscript. Considering that the bulk of the data analysis directly involves the concept of microbial generalists or specialists.

Another issue with the paper is the use of genome-resolved metagenomics to differentiate the metabolic traits of habitat generalists and specialists. They show “The habitat generalist MAGs exhibited higher metabolic flexibility (encoding on average 8.8 and 260 7.5 signature metabolic genes in the Antarctic and Swiss datasets respectively) 261 compared to the habitat specialist MAGs (6.5 and 6.2 respectively)”. However, they do not clarify how the signature genes were selected. It would greatly help the manuscript if the authors would add a more comprehensive description of these signature genes.

If the authors would work to clarify these methodological details, and assuming all methods are sound, this work would be more likely to be reproducible.

Reviewer #2

(Remarks to the Author)

Glacier recession is creating novel terrestrial habitats at vast scale this century, and how these ecosystems are formed provides fundamental insights in ecology. Ricci et al. use samples from two very different glacier forefields to identify the strategies used by microbes to colonize glacier forefields, concluding metabolic flexibility including the metabolism of atmospheric trace gases, plays a central role.

I have read the manuscript carefully and simply can not identify any reason why this manuscript should not be published as-is. It represents an excellent piece of work which links together amplicon, metagenomic, physical/chemical analyses, bioinformatic and community modelling, and gas flux incubations to draw robust conclusions. The team should be congratulated on their work.

If there is an area for improvement, I would recommend the authors make clear that they have selected two very different glacial systems (a very truncated maritime Antarctic glacier forefield and an alpine glacier forefield) which could be expected to be highly contrasting, and yet they find key commonalities in how they are colonized by microbes. While it is impossible to sample all glacier forefields, this approach makes a meaningful effort at generalizing the trends observed, but it remains a

small sample. Some lines reflecting on the rationale and limitations of this approach in the discussion would be beneficial, but not essential.

Reviewer #3

(Remarks to the Author)

Version 1:

Reviewer comments:

Reviewer #1

(Remarks to the Author)

The revised manuscript addresses very clearly and completely the concerns that I had expressed in my previous review. I am pleased to say that this manuscript is now acceptable in its current form.

**Responses to Reviewer #1 (Remarks to the Author):**

The draft article by Ricci et al. provides excellent points that have implications for how ecosystems
are formed. The authors investigated the functional basis of microbial colonization in the Antarctic
and an alpine Swiss retreating glacier by integrating data from genome resolved metagenomics,
quantitative ecology, and biogeochemical data. The paper provides interesting data and concepts
for how generalists and specialist microbial communities colonize during succession. The paper is
of high importance and relevance to this journal. It is also exceptionally well written and
communicated.

**We thank the reviewer for their positive comments regarding the importance, quality, and**
**presentation of the work.**

The major concern with the manuscript is regarding the different indices to determine habitat
specializations (Relative Habitat Use, RHU). The authors used the coefficient of variance to
determine generalist and specialist species, using both amplicon and metagenomics sequences.
We strongly recommend the authors provide a more concrete basis and justification, in addition to
more relevant citations why they used the first quartile and fourth quartile as threshold to determine
species specialization, it would be helpful for the manuscript. Considering that the bulk of the data
analysis directly involves the concept of microbial generalists or specialists.

**We agree with the reviewer that this needs to be clearly rationalised. A wide range of**
**approaches have been used to classify habitat generalism in ecology and there remains no**
**clear consensus on how best to do this. In our case, we used the ‘habitat specialisation**
**index’, i.e. coefficient of variance of taxon relative abundance, as originally defined in the**
**well-cited avian ecology paper by Julliard et al., Ecology Letters 2006. This simple metric**
**allows us to quantify the relative variability of a taxon across environments in a**
**standardised way in both amplicon and MAG datasets. This metric can also be convincingly**
**applied to microbial data, for example as in Chen et al., ISME J 2020. We are strong**
**advocates for applying macroecological theory and methods to microbial systems.**

**As the reviewer indicated, we categorised relative habitat ‘generalists’, ‘intermediates’, and**
**‘specialists’ based as the lower quartile, middle quartiles, and upper quartiles respectively**
**of habitat specialisation. While habitat generalisation is a continuous distribution, focusing**
**on the quartiles enables us to focus on the taxa representing the most ecologically distinct**
**ends of the spectrum in a quantitatively defined way. Similar quartile-based approaches**
**have been used both in macroecology and microbial ecology studies to discriminate**
**specialists and generalists, for example Oplante et al., Science 2024, though many other**
**approaches have also been used by applying other quantiles or z-scores instead. In this**
**case, using quartiles ensures that we retain sufficient taxa for comparison while**
**maintaining contrast between groups for downstream comparisons; more extreme**
**thresholds (e.g., deciles) would reduce group sizes and statistical power, while broader**
**cutoffs could dilute ecological signal. This is now reflected in the revised methods section:**

**L849-875: “Habitat specialization indices: The degree of habitat specialization of each MAG**
**(i.e. whether they were relative ‘habitat generalists’ or ‘habitat specialists’) was calculated**
**based on the coefficient of variance in the metagenomes (i.e. relative mean divided by**
**relative standard deviation calculated by read mapping). This metric, termed the ‘habitat**
**specialization index’, was originally proposed in macroecology^{61,114} and has since been**
**effectively applied in microbial systems²¹; it quantifies niche breadth in a way that is simple,**
**continuous, and extendible across data types (e.g., amplicon and metagenomic datasets),**

supporting a macroecological framework in microbial ecology. The community-wide mean
and standard deviation specialization index was 1.459 ± 0.621 for the Antarctic MAGs and
1.302 ± 0.792 for the Swiss MAGs. Relative ‘habitat generalists’ were defined as MAGs with
a specialization index (i.e. coefficient of variance) below the first quartile (below 0.9886 for
Antarctic MAGs, 0.6652 for Swiss MAGs), relative ‘habitat specialists’ were defined as MAGs
with a specialization index in the fourth quartile (above 1.8751 for Antarctic MAGs, 1.7778
for Swiss MAGs), and relative ‘habitat intermediates’ are within the interquartile range.
Inspired by previous studies in microbial ecology¹¹⁵, these quartile-based thresholds enable
discrimination of ecologically distinct groups within a continuous distribution, while
ensuring sufficient taxa are contained within each group for meaningful statistical
comparisons. In contrast, more extreme thresholds would reduce statistical power, while
broader cutoffs would dilute ecological signal. To disentangle the effects of soil age and
depth for the Swiss samples, habitat specialization index was only calculated based on read
mapping to the surface metagenomes. This quartile-based specialization index was also
used to classify taxa based on the 16S rRNA gene amplicon sequencing data. Specifically,
the habitat specialization indices of each genus and phylum with an average relative
abundance exceeding 0.005% was also calculated, based on their coefficient of variance;
again, habitat generalists and specialists defined as taxa with specialization indices below
the first quartile and above the fourth quartile respectively.”

**As well as the following note in the Results section:**

**L246-248: “this quartile-based classification enabled us to quantitatively discriminate**
**ecologically distinct taxa while ensuring sufficient MAGs were contained within each group**
**to enable robust comparison of their distributions and capabilities.”**

**Furthermore, we now explicitly discuss the limitations of this method and justify its**
**application in the context of our dataset, where a continuous gradient of habitat preference**
**is observed.**

**L567-571: “It should be noted that, whereas habitat generalism is a continuous distribution,**
**this study primarily focused on the upper and lower quartiles of this distribution; future**
**studies should explore the distributions and capabilities of microbes across the full**
**spectrum of generalism by applying continuous analyses or finer percentile-based**
**classifications.”**

Another issue with the paper is the use of genome-resolved metagenomics to differentiate the
metabolic traits of habitat generalists and specialists. They show “The habitat generalist MAGs
exhibited higher metabolic flexibility (encoding on average 8.8 and 260 7.5 signature metabolic
genes in the Antarctic and Swiss datasets respectively) 261 compared to the habitat specialist
MAGs (6.5 and 6.2 respectively)”. However, they do not clarify how the signature genes were
selected. It would greatly help the manuscript if the authors would add a more comprehensive
description of these signature genes.

**The metabolic genes selected encode essential (typically catalytic) subunit proteins for all**
**major energy, carbon, and nitrogen acquisition processes. They are an extremely carefully**
**curated in-house dataset that capture the breadth of metabolic processes while avoiding**
**the extensive false positives and negatives associated with many similar approaches (e.g.**
**KEGG). This enables us to reliably quantify the levels of metabolic genes in short reads,**
**compared to single-copy universal marker genes, as well as analyse their distribution in**
**MAGs. The original manuscript notes the following in the results section: “The metabolic**

**versatility of each MAG was inferred based on whether they encoded 56 signature**
**metabolic genes, including the primary dehydrogenases for organotrophy and lithotrophy,**
**the terminal reductases for aerobic and anaerobic respiration, and the signature enzymes**
**for carbon fixation, photosystem- and rhodopsin-based phototrophy, and nitrogen fixation.”**
**In the revised manuscript, we have given more details about our approach in the methods**
**section:**

**“Metagenomic short read analysis: To estimate the metabolic capability of the soil**
**communities, quality-filtered short reads were searched against custom protein databases**
**of 56 metabolic marker genes**
**([https://bridges.monash.edu/collections/Greening_lab_metabolic_marker_gene_databases/](https://bridges.monash.edu/collections/Greening_lab_metabolic_marker_gene_databases/5230745)**
**5230745). These databases contain the amino acid sequences of an essential (usually**
**catalytic) subunit of a key enzyme for each of the major pathways of energy, carbon, and**
**nitrogen acquisition. They include enzymes catalysing the six pathways of carbon fixation,**
**photosystem- and rhodopsin-based phototrophy, aerobic and anaerobic respiration, and**
**nitrogen, sulfur, iron, hydrogen, carbon monoxide, and methane cycling, as listed in Table**
**S3. As previously described, these databases were carefully manually constructed to**
**encompass the known diversity of each enzyme^{91–93}. DIAMOND v.0.9.31 searches⁹⁴ were**
**conducted with a query coverage of 80% and identity threshold of 50%, except for *rho***
**(40%), *nuoF*, group 4 [NiFe]-hydrogenases, *mmoX*, [FeFe]-hydrogenases, *coxL*, *amoA*, *nxA*,**
***rbcL* (all 60%), *psaA* (80%), *psbA*, *isoA*, *atpA*, *aro*, *ygfK* (70%), and *hbsT* (75%). These**
**carefully curated databases and search criteria enable metagenomic short reads encoding**
**each metabolic gene to be accurately quantified. The average gene copy per organism is**
**calculated by dividing the read counts for each gene (in reads per kilobase million, RPKM)**
**against the read count for universal single-copy ribosomal protein genes in RPKM), as**
**previously described^{19,50,95}.”**

**In addition, we have added an additional table in Table S3 where we list the proteins, their**
**functions, and their search criteria.**

If the authors would work to clarify these methodological details, and assuming all methods are
sound, this work would be more likely to be reproducible.

**We thank the reviewer. Our approaches and rationale should now be much more clear.**

**Responses to Reviewer #2 (Remarks to the Author):**

Glacier recession is creating novel terrestrial habitats at vast scale this century, and how these
ecosystems are formed provides fundamental insights in ecology. Ricci et al. use samples from two
very different glacier forefields to identify the strategies used by microbes to colonize glacier
forefields, concluding metabolic flexibility including the metabolism of atmospheric trace gases,
plays a central role.

I have read the manuscript carefully and simply can not identify any reason why this manuscript
should not be published as-is. It represents an excellent piece of work which links together
amplicon, metagenomic, physical/chemical analyses, bioinformatic and community modelling, and
gas flux incubations to draw robust conclusions. The team should be congratulated on their work.

**We greatly appreciate the reviewer's comments. This manuscript was a tremendous amount**
**of work and we are pleased with the results. We are glad it has received such glowing**
**responses.**

If there is an area for improvement, I would recommend the authors make clear that they have
selected two very different glacial systems (a very truncated maritime Antarctic glacier forefield and
an alpine glacier forefield) which could be expected to be highly contrasting, and yet they find key
commonalities in how they are colonized by microbes. While it is impossible to sample all glacier
forefields, this approach makes a meaningful effort at generalizing the trends observed, but it
remains a small sample. Some lines reflecting on the rationale and limitations of this approach in
the discussion would be beneficial, but not essential.

**We agree with the reviewer. The manuscript does capture the many commonalities and key**
**differences in the patterns observed between these two highly contrasting glaciers,**
**especially through the synthesis in the Conclusions. However, it's unclear to what extent**
**these features apply to the many other glacial ecosystems worldwide. We have therefore**
**noted the following in the Conclusions:**

**L574-578: "In addition, while we observed similar findings across two divergent glacial**
**forelands, it is unclear to what extent what extent these findings predict colonisation in**
**other forelands, as well as primary and secondary succession in other ecosystems (e.g.**
**volcanic soils, meteorites, post-fire recovery)."**

**Responses to Reviewer #3 (Remarks to the Author):**

I co-reviewed this manuscript with one of the reviewers who provided the listed reports. This is part
of the Nature Communications initiative to facilitate training in peer review and to provide
appropriate recognition for Early Career Researchers who co-review manuscripts.

**We thank the early-career researcher for assisting in this peer review process.**